# New PEPTIR-2.0 Peptide Designed for Use as Recognition Element in Electrochemical Biosensors with Improved Specificity towards *E. coli* O157:H7

**DOI:** 10.3390/molecules27092704

**Published:** 2022-04-22

**Authors:** Jose Luis Ropero-Vega, Joshua Felipe Redondo-Ortega, Juliana Paola Rodríguez-Caicedo, Paola Rondón-Villarreal, Johanna Marcela Flórez-Castillo

**Affiliations:** 1Universidad de Santander, Facultad de Ciencias Naturales, Ciencias Básicas y Aplicadas Para la Sostenibilidad—CIBAS, Calle 70 No. 55-210, Santander, Bucaramanga C.P. 680003, Colombia; hugofelipez@hotmail.com (J.F.R.-O.); julianapao6@gmail.com (J.P.R.-C.); johanna.florez@udes.edu.co (J.M.F.-C.); 2Universidad de Santander, Facultad de Ciencias Médicas y de la Salud, Instituto de Investigación Masira, Calle 70 No. 55-210, Santander, Bucaramanga C.P. 680003, Colombia; diseno.molecular@udes.edu.co

**Keywords:** PEPTIR-1.0, bioinformatics tools, pathogen, water, electrochemical impedance spectroscopy, high specificity

## Abstract

The detection of pathogens through alternative methodologies based on electrochemical biosensors is being studied. These devices exhibit remarkable properties, such as simplicity, specificity, and high sensitivity in monitoring pathogens. However, it is necessary to continue conducting studies that adequately improve these characteristics, especially the recognition molecule. This work aims to design and evaluate a new peptide, named PEPTIR-2.0, as a recognition molecule in electrochemical biosensors to detect *E. coli* O157:H7 in water. PEPTIR-2.0 was obtained from modifications of the PEPTIR-1.0 peptide sequence, which was previously reported and exhibited excellent properties for detecting and quantifying this pathogenic microorganism. PEPTIR-1.0 is a peptide analogous to the TIR (Translocated Intimin Receptor) protein capable of interacting with the Intimin outer membrane. The basis of this study was to obtain, by using bioinformatics tools, a molecule analogous to PEPTIR-1.0 that maintains its three-dimensional structure but increases the hydrophobic interactions between it and Intimin, since these intermolecular forces are the predominant ones. The designed PEPTIR-2.0 peptide was immobilized on screen-printed electrodes modified with gold nanoparticles. The detection capacity of *E. coli* O157:H7 in water was evaluated using electrochemical impedance spectroscopy in the presence of other microorganisms, such as *P. aeruginosa*, *S. aureus*, and non-pathogenic *E. coli*. The results showed that PEPTIR-2.0 confers remarkable specificity to the biosensor towards detecting *E. coli*, even higher than PEPTIR-1.0.

## 1. Introduction

Methodologies based on electrochemical biosensors have been studied due to their remarkable properties to control and monitor pathogens, such as *E. coli* O157:H7 [1,2,3,4,5,6]. Electrochemical biosensors are instruments that offer advantages, such as simplicity, sensitivity, and specificity, which make them an efficient alternative for the detection and monitoring of pathogens. These devices are composed of an element of biological nature and a physicochemical transducer [7,8]. The biological element is the component that confers specificity to the biosensor. Any biological structure that has this capacity is usable. Enzymes, antibodies, nucleic acids (DNA or RNA), subcellular organelles, fine sections of cellular tissues, membrane receptors, and biomimetic components such as peptides are commonly used. On the other hand, the transducers are generally composed of conductive supports based on carbon nanostructures coupled with metallic nanoparticles, which increase the surface area and amplify the electroactivity of the biosensor. In addition, the transducers convert the biochemical activity of the recognition elements into a measurable signal [9,10,11,12].

Peptide-based electrochemical biosensors are probably the most versatile systems due to the tunable physicochemical properties of these molecules [1,13]. They have the same protein building block and thus can be folded into compact structural motifs that shape nano-sized architectures, such as monolayers, tubes, bilayers, strips, micelles, and fibers. Peptides can carry intermolecular, non-covalent, electrostatic, aromatic ring stacking, hydrogen bonding, hydrophobic, and van der Waals interactions with molecular targets. They can be designed to replace domains of interest of macromolecules, such as antibodies, enzyme substrates, or protein receptor binding sites. Therefore, it is noteworthy that the peptides designed in an analogous way to receptors can be specific molecules in molecular recognition that allow the detection of bacteria through interaction with their membrane proteins, also providing information on the presence of structures related to virulence or pathogenic mechanisms of great importance [14,15,16].

We have recently reported the bioinformatic design of a new peptide named PEPTIR-1.0, a molecule analogous to the TIR (Translocated Intimin Receptor) protein which is a receptor for the Intimin membrane protein characteristic of *E. coli* O157:H7 [17]. These proteins are encoded by chromosomal genes (TIR gene and eaeA gene) of enterohemorrhagic and enteropathogenic strains of *Escherichia coli*. Therefore, PEPTIR-1.0 was evaluated as a recognition element in an electrochemical biosensor to detect this pathogenic microorganism. The results showed that this peptide confers adequate specificity to the biosensor to monitor this pathogen in an aqueous matrix.

The interaction of PEPTIR-1.0 with the Intimin membrane protein occurs mainly through hydrophobic interactions and, to a lesser extent, hydrogen bonds. In this sense, the aim of this work was the bioinformatic design of a new peptide by modifying the PEPTIR-1.0 sequence, maintaining its three-dimensional structure but improving hydrophobic interactions with the Intimin protein. This new peptide, named PEPTIR-2.0, was used as a recognition element in a biosensor based on screen-printed electrodes modified with gold nanoparticles (AuNPs). The limits of detection and quantification of *E. coli* O157: H7 in water were evaluated. In addition, the specificity of this new biosensor based on PEPTIR-2.0 was studied in the presence of other microorganisms, such as *Pseudomonas aeruginosa*, *Staphylococcus aureus*, and non-pathogenic *E. coli*, and the obtained results were highly satisfactory.

## 2. Results and Discussion

### 2.1. Modeling of Sequences Analogous to the PEPTIR Molecule

From the modifications made to the original 20 amino acid sequence of the PEPTIR-1.0 molecule (QKVNIDELGNAIPSGVLKDD), 22 different sequences were obtained (see Appendix A). For each one, the 3D structure was obtained using the PEP-FOLD program. Then, the three closest analogous to the PEPTIR-1.0 molecule, in terms of the RMSD (root mean square deviation) given by the Pymol program, were selected to perform molecular docking simulations. The selected analogous peptides with their corresponding RMSD value is shown in Table 1.

Figure 1 shows the 3D structure of the original PEPTIR-1.0 model (red) and analogous peptides 1, 2, and 3 (blue, green, and magenta, respectively) selected as the best option for molecular coupling assays with the Intimin protein.

### 2.2. Molecular Docking between the Newly Designed Molecule and the INTIMIN Protein

The interaction models obtained using the FlexPepDock program for the three analogous peptides selected together with the original structure of the Intimin protein (chain A obtained from PDB file ID: 2ZQK) are shown in Figure 2.

Table 2 shows the binding affinity, dissociation constant, interface energy, and the number of interactions obtained by molecular docking between the three analogous peptides and the structure of the Intimin protein using the FlexPepDock and Prodigy Haddock programs. The results of the original TIR protein (2ZQK model) and PEPTIR-1.0 are included for comparison purposes.

Analog 2 presents the lowest ΔG, K_d_, and I_SC values. In addition, this model has a higher number of intermolecular interactions that occur at the interface of the peptide and Intimin protein, with a total of 63 interfacial contacts, being even higher compared to the PEPTIR-1.0 and the original chain of the TIR protein, which are 52 and 62 interactions, respectively.

These results are promising, and it can be expected that the new peptide model obtained from the sequence QKVNIAELGNAIPSGVLKDD shows a great capacity to interact appropriately with the Intimin protein. Therefore, this peptide (named PEPTIR-2.0) was selected as the recognition molecule to prepare the new electrochemical biosensor.

### 2.3. Preparation and Evaluation of PEPTIR Based Biosensors

Biosensors were prepared using carbon-based screen-printed electrodes, according to the scheme shown in Figure 3. First, gold nanoparticles were deposited on the working electrode by reduction of the HAuCl_4_ gold precursor by chronoamperometry at −0.05 V vs. Ag/AgCl at 100 s (Figure 3a,b). Subsequently, the immobilization of the PEPTIR peptides was carried out (Figure 3c), followed by the evaluation in the detection of microorganisms (Figure 3d).

Scanning electron microscopy analysis showed that the biosensors have a homogeneous distribution of gold nanoparticles on the surface. The size of the nanoparticles is around 50 ± 12 nm.

In this work, the effect of peptides concentration in the immobilization solution on the electrochemical response of the biosensor towards *E. coli* O157:H7 was evaluated. In addition, the monitoring of changes in the electrochemical response of the biosensors was carried out using SWV and EIS. In all cases, the electrochemical response follows a behavior like the one presented in Figure 4.

The results of the other peptide concentrations are found in the Appendix A. The monitoring of the changes in the electrochemical properties during the preparation of the biosensors and their evaluation in detecting the bacteria was carried out using the hexacyanoferrate (II/III) redox probe [9,18]. The screen-printed electrodes present a relatively low charge transfer property that is reflected in a maximum current of around 150 µA in SWV and high impedance values according to the EIS results (black curves in both techniques). The deposition of gold nanoparticles leads to a considerable improvement in the charge transfer properties of the electrodes, evidenced by an increase in the maximum current in SWV and a decrease in impedance in the Nyquist diagrams (red curves in both techniques). Finally, the immobilization of the peptides leads to a slight limitation in the transfer of charge on the surface of the electrodes, evidenced by a slight decrease in the maximum current in SWV and a slight increase in the impedance in EIS (blue curves in both techniques).

As expected, the interaction of *E. coli* with the biosensors limits the charge transfer processes mediated by the electrochemical redox probe. In the case of SWV analyses, these limitations are reflected in a decrease in the intensity of the maximum current peak as the concentration of the microorganism increases (Figure 4, left).

Electrochemical impedance spectroscopy allows a more detailed analysis of these changes through Nyquist diagrams (Figure 4, center). The figure inset shows the equivalent circuit that fits appropriately with the Nyquist diagrams, obtained both in the preparation of the biosensor and in the evaluation of the detection of *E. coli*. This circuit, which is well-known as the Randles circuit, represents the typical resistance to the solution (R_sol_), a constant phase element (CPE) associated with the electrical double layer that forms on the surface of the biosensor, and the resistance to charge transfer (R_ct_) that is strongly affected during the use of the biosensor. Finally, the Warburg impedance (W) is usually found in this type of system, as it is associated with diffusive processes on the electrode [5,19,20,21,22,23].

In the case of impedimetric electrochemical biosensors, the resistance to charge transfer is expected to increase proportionally with the increase in the target of interest [5]. The results show that this behavior is linear for PEPTIR-1.0 electrodes prepared with a peptide concentration of 10 nM, while this linearity is observed for PEPTIR-2.0 biosensors prepared with 5 nM peptide (Figure 4, right). The results for all the concentrations evaluated can be found in the Appendix A.

The detection and quantification limits for each of the prepared biosensors were calculated from the equation *k**·S_bl_*/*m*, where *k* is 3 for the detection limit at a confidence level of 98.3% and 10 for the quantification limit; *S_bl_* is the standard deviation of the blank and *m* is the slope of the line. The parameters obtained from each of the biosensors are shown in Table 3.

In the calibration curves, the variability or standard deviation in the biosensor response in each of the concentrations of *E. coli* evaluated is greater for PEPTIR-1.0 than for PEPTIR-2.0. Moreover, the line slope is slightly higher for the second than for the first. Finally, the detection and quantification limits of the biosensor based on PEPTIR-2.0 are lower, as expected according to the bioinformatics results.

The specificity of the biosensors towards the detection of *E. coli* O157:H7 was evaluated in the presence of other microorganisms, such as *Pseudomonas aeruginosa*, *Staphylococcus aureus*, and non-pathogenic *E. coli*. The results are shown in Figure 5.

In both cases, it was observed that the biosensors do not show a response towards *P. aeruginosa* (gray bars). This follows because the biosensor response towards this microorganism is like the detection blank (0 CFU/mL, 0.9% *w*/*w* of NaCl). In the case of PEPTIR-1.0, the biosensor can respond to the presence of non-pathogenic *E. coli* and, to a lesser extent, *S. aureus*, especially at high concentrations of these microorganisms. For its part, PEPTIR-2.0 exhibits a marked specificity towards *E. coli* O157:H7, even at a concentration of 200 CFU/mL of all microorganisms. In fact, the response of PEPTIR-2.0 towards *P. aeruginosa*, *S. aureus*, and non-pathogenic *E. coli* at concentrations of 200 CFU/mL is significantly similar to the one exhibited by the biosensor in the detection blank.

The higher sensitivity and specificity that PEPTIR-2.0 exhibits compared to PEPTIR-1.0 towards detecting *E. coli* O157:H7 can be attributed to the better molecular interaction properties that the former possesses. This represents a considerable advance in developing rapid detection devices for pathogenic microorganisms in water based on electrochemical biosensors.

## 3. Materials and Methods

### 3.1. Reagents, Materials, and Instruments

All reagents were used as received without further purification: potassium chloride (KCl ≥99%, Sigma Aldrich, Burlington, MA, USA), potassium hexacyanoferrate (II) (K_4_[Fe(CN)_6_] × 3H_2_O, ≥98%, Merck), potassium hexacyanoferrate (III) (K_3_[Fe(CN)_6_], ≥99%, Merck), gold (III) chloride hydrate (HAuCl_4_ × 3H_2_O, ≥99%, Sigma Aldrich), LB-Agar (Merck). The peptides, PEPTIR-1.0 (sequence QKVNIDELGNAIPSGVLKDD-NH_2_) and PEPTIR-2.0 (sequence QKVNIAELGNAIPSGVLKDD-NH_2_), were synthesized by Biomatik^®^ (Wilmington, DE, USA) with a purity of >95%. A cysteine was included in the N-terminal region of the chains. The bacterial strains used were the references *Escherichia coli* O157:H7 (ATCC 43895), non-pathogenic *Escherichia coli* (ATCC 25922), *Staphylococcus aureus* (ATCC 25923), and *Pseudomonas aeruginosa* (ATCC 27853).

Screen-printed electrodes were supplied by Italsens and consisted of a working carbon electrode (7.07 mm^2^), an auxiliary carbon electrode, and a silver/silver chloride (Ag/AgCl) reference electrode. Electrochemical measurements of the biosensor were performed on a potentiostat/galvanostat VersaSTAT 3 (Princeton Applied Research, AMETEK, Berwyn, PA, USA) controlled by Versastudio (v. 2.60.6.) software.

### 3.2. Modeling of Sequences Analogous to the PEPTIR Molecule

From the amino acid sequence of the PEPTIR-1.0 molecule (QKVNIDELGNAIPSGVLKDD) and depending on the predominance of hydrophobic interactions between this ligand and the Intimin protein, the substitutions in positions 6, 7, and 18 can be replaced by the amino acids: lysine, valine, glycine, alanine, methionine, tryptophan, and leucine, to increase the affinity with the Intimin protein.

Subsequently, the 3D structures of the selected sequences were obtained with the PEP-FOLD program [24], an online system based on the concept of a structural alphabet that describes the possible conformations for fragments of 4 consecutive residues. Finally, the predicted structures were analyzed using the RMSD values given by the Pymol program [25] when compared with the PEPTIR-1.0 structure. The models with the smallest structural differences were selected for the molecular docking simulations.

### 3.3. Molecular Docking between the Newly Designed Molecule and the Intimin Protein

The PDB entry 2ZQK from the RCSB PDB database corresponds to the interaction model of the Intimin and TIR *E. coli* O157:H7 proteins proposed by Ma, Y. et al. [26]. From this model, we selected chain A, which corresponds to the 3D structure of the Intimin protein. This chain, together with the previously chosen models obtained with the PEP-FOLD program, were used to carry out molecular docking simulations, which refers to the study of the capacity of interaction between a ligand (peptide models—Chain N) and a receptor (Intimin Protein—Chain A). The simulations were carried out using Rosetta software. Through its FlexPepDock protocol, this program allows the prediction of models between peptides and proteins by iteratively optimizing the peptide skeleton and its rigid orientation of the body relative to the receptor protein [26]. In addition, the values obtained with the FlexPepDock and Prodigy program (interface energy, binding affinity, dissociation constant, number of interactions, among others) were compared between the different models evaluated, and the docking results using the original PEPTIR-1.0 molecular model as a ligand. The obtained results defined the new recognition molecule used in the electrochemical biosensor.

### 3.4. Preparation of the Electrochemical Biosensors

Biosensors were prepared through the methodology previously published [17]. Briefly, screen-printed electrodes (SPE) were modified with gold nanoparticles (AuNPs) by electrochemical reduction by chronoamperometry (−0.05 V vs. Ag/AgCl, 100 s) of a solution of HAuCl_4_ (1.0 mM in 0.5 M of H_2_SO_4_). Then, the PEPTIR (1.0 and 2.0) peptides were immobilized by self-assembling these molecules through an Au–S bond between the thiol group of terminal cysteine of the peptides and AuNPs. For this, 10 µL of an aqueous solution of PEPTIR were placed on the AuNPs-modified working electrode of the SPE and maintained at 25 °C overnight. Finally, the electrode was rinsed with Type-I water. The concentration of PEPTIR for immobilization was evaluated in the range of 5 to 100 nM.

The surface characteristics of the biosensors were studied by scanning electron microscopy (SEM) using a Quanta Field Emission Gun microscope (Model 650) operated at 15.0 kV. Images were obtained in backscattered electron mode.

### 3.5. Electrochemical Characterization of the Biosensors

The screen-printed electrodes were characterized throughout the biosensor preparation process. Cyclic voltammetry (CV) was made with a scan potential between −0.4 and +0.8 V vs. Ag/AgCl at a scan rate of 25 mV/s. Square wave voltammetry (SWV) measurements were made using a scan potential between −0.4 and +0.8 V vs. Ag/AgCl, 75 mV of pulse height, 2.5 mV of step height and 10 Hz. Electrochemical impedance spectroscopy (EIS) measurements were performed in a frequency scan between 50,000 to 1 Hz, using 10 mV of AC potential and under open circuit potential (DC). Voltammetry measurements were made by using an electrolytic solution of 10.0 mM of hexacyanoferrate (II/III) in 0.1 M of KCl (supporting electrolyte), while EIS measurements were made at 5.0 mM of hexacyanoferrate (II/III) in the same concentration of supporting electrolyte.

### 3.6. Detection of E. coli Cells Using the Electrochemical Biosensors

For the electrochemical detection of *E. coli* O157:H7 ATCC^®^ 43895, a 5 × 10^5^ CFU/mL stock solution was established in 10 mM PBS at pH 7.4. This was done after standardizing the concentration of the bacterium with the standard method of optical density vs. colony count (surface count) from liquid cultures with 18 to 24 h of growth at 37 °C and 180 rpm in a Luria-Bertani (LB) medium. Then, 7.0 mL of the solutions with *E. coli* at the known concentration were exposed to the electrochemical biosensor for an incubation time of 30 min. Electrochemical measurements were made after the respective washes with ultra-pure water. The selectivity of the biosensor was evaluated in the presence of other microorganisms, such as non-pathogenic *E. coli*, *P. aeruginosa*, and *S. aureus*. All microorganism detection measurements were performed in triplicate at each concentration evaluated.

## 4. Conclusions

We have obtained a new peptide named PEPTIR-2.0 by modifying the sequence of the PEPTIR-1.0 peptide. This modification consisted of substituting an aspartic acid residue with alanine at position 6 of PEPTIR-1.0. This single change significantly modifies the peptide-Intimin interaction, increasing the number of interactions between both structures and decreasing the free energy and constant dissociation values.

The use of PEPTIR-2.0, as a recognition element in electrochemical biosensors based on gold nanoparticles-modified screen-printed electrodes, exhibits a higher specificity towards *E. coli* O157:H7 in an aqueous matrix in comparison to PEPTIR-1.0.

## Figures and Tables

**Figure 1 molecules-27-02704-f001:**
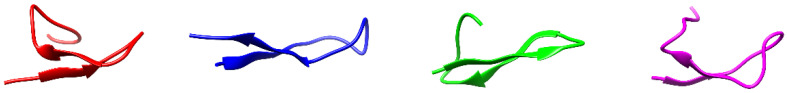
Three-dimensional structure of the sequences: QKVNIDELGNAIPSGVLKDD (PEPTIR-1.0, **Red**), and the analogous 1 (QKVNIAELGNAIPSGVLKDD, **Blue**), 2 (QKVNIAELGNAIPSGVLKDD, **Green**), and 3 (QKVNIMMLGNAIPSGVLMDD, **Magenta**).

**Figure 2 molecules-27-02704-f002:**
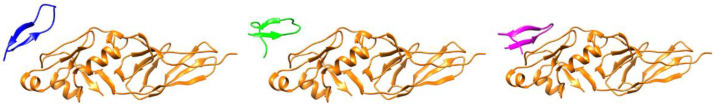
The obtained models with the FlexPepDock program for the analogous peptides 1 (**Blue**), 2 (**Green**), and 3 (**Magenta**) interact with the structure of the Intimin protein.

**Figure 3 molecules-27-02704-f003:**
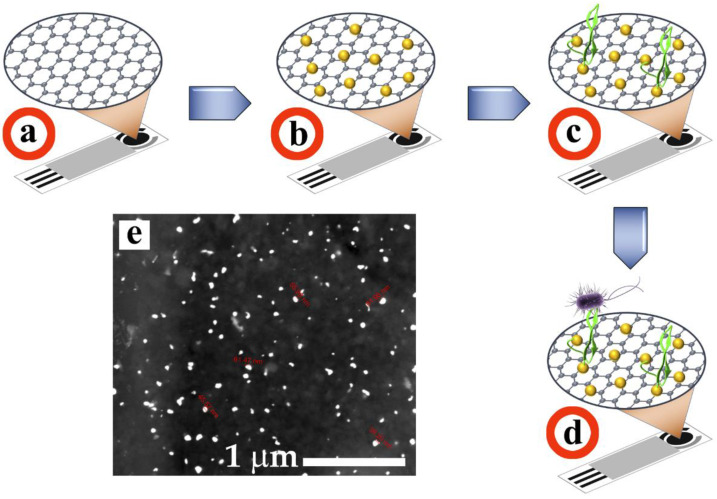
Schematic representation of the preparation and evaluation of the biosensors step by step. Screen-printed carbon-based electrodes (**a**) were modified with gold nanoparticles (**b**), followed by PEPTIR immobilization (**c**) and detection tests (**d**). The micrograph (**e**) was obtained in backscattered electron mode.

**Figure 4 molecules-27-02704-f004:**
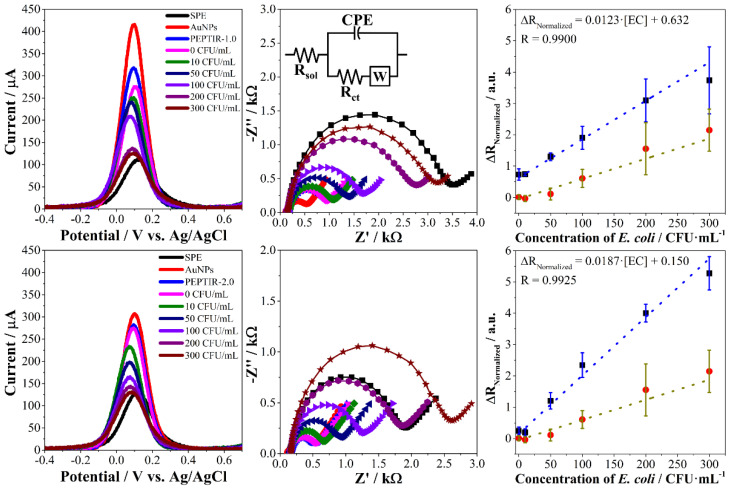
Square wave voltammograms (**left**), Nyquist plots (**center**), and calibration curves (**right**) of the biosensors prepared with 10 nM of PEPTIR-1.0 (**top**) and 5 nM of PEPTIR-2.0 (**bottom**) in the detection of *E. coli* O157:H7. The red points in the curves on the right correspond to the calibration curve of the biosensor blank (without using PEPTIR).

**Figure 5 molecules-27-02704-f005:**
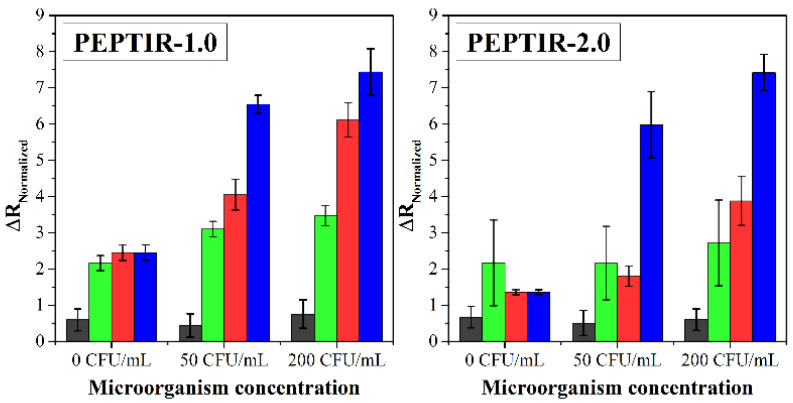
Effect of different concentrations of bacteria on the electrochemical response of the biosensors prepared. Aqueous matrix doped with *Pseudomonas aeruginosa* (gray bars), *Staphylococcus aureus* (green bars), *E. coli* ATCC 25922 (red bars), and *E. coli* O157:H7 (blue bars).

**Table 1 molecules-27-02704-t001:** RMSD values associated with the three analogous peptides selected.

Analogous Peptides	Sequence	RMSD
PEPTIR-1.0	QKVNIDELGNAIPSGVLKDD	4.25
1	QKVNILLLGNAIPSGVLLDD	5.48
2	QKVNIAELGNAIPSGVLKDD	3.50
3	QKVNIMMLGNAIPSGVLMDD	4.86

The amino acids that change from the original PEPTIR-1.0 sequence appear in red type.

**Table 2 molecules-27-02704-t002:** Interaction parameters between the analogous peptides and the Intimin protein, obtained with the FlexPepDock and Prodigy Haddock programs. ΔG: binding affinity, K_d_: dissociation constant, I_SC: interface energy.

Model	ΔG (kcal/mol)	K_d_ (mol/L)	I_SC	Interactions
TIR protein	−11.50	3.7 × 10^−9^	-	62
PEPTIR-1.0	−10.9	9.6 × 10^−9^	−18.106	52
1	−10.7	1.4 × 10^−8^	−18.790	51
2	−13.1	2.6 × 10^−10^	−24.329	63
3	−11.8	2.0 × 10^−9^	−10.900	41

**Table 3 molecules-27-02704-t003:** Analytical parameters of the biosensors.

Biosensor	Limit of Detection (CFU/mL)	Limit of Quantification (CFU/mL)	Calibration Sensitivity (1/CFU·mL^−1^)
PEPTIR-1.0	44	146	0.0123
PEPTIR-2.0	19	65	0.0187

## Data Availability

The data presented in this study are available in Appendix A. Samples of the PEPTIR-1.0 and PEPTIR-2.0 are available from the authors.

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
