# Peer review of "New PEPTIR-2.0 Peptide Designed for Use as Recognition Element in Electrochemical Biosensors with Improved Specificity towards E. coli O157:H7"

_molecules, 2022, doi:10.3390/molecules27092704_

Round 1

Reviewer 1 Report

Dear Editor

First, thank you for inviting me to review this manuscript. In this cover letter. I have included my comments and suggestions

Manuscript Number: molecules-1685116

By modifying the sequence of PEPTIR-1.0, authors have derived a new peptide named PEPTIR-2.0. PEPTIR-1.0 was modified by replacing an aspartic acid residue with an alanine at position 6. By modifying this particular interaction, the number of interactions between the peptide and the Intimin increases and the free energy and constant dissociation values decrease. Authors also demonstrated that based on gold nanoparticles-modified screen-printed electrodes, PEPTIR-2.0 is used as a recognition element in electrochemical biosensors. In comparison with PEPTIR-1.0, it has a higher specificity for E.coli in an aqueous matrix.   

The conclusions are well supported by the results. Overall, the presentation of this paper is great. I would like to recommend the publication of this work in this journal after some issues are well modified or clarified.

  • There are many errors in English writing in manuscripts. Perhaps the author should proofread it.
  • Little has been studied about the electrode. It is recommended that the author characterizes and discusses the electode they prepare. It helps readers understand the condition more and repeat the experiments.

Reviewer 2 Report

This is an elegant study concerning a very important issue, the easy detection of E. coli O157:H7

The results support the conclusions and the importance of this issue is enormous. I really consider this a very important study that deserve to be published and I only have a very small comment on it:

Pag 4: Line 123 where it is written “In this work, the effect of the concentration of the peptides in…” I would have written “In this work, the effect of peptides concentration in…”
